# PTSD Symptoms and Coping with COVID-19 Pandemic among Treatment-Seeking Veterans: Prospective Cohort Study

**DOI:** 10.3390/jcm11102715

**Published:** 2022-05-11

**Authors:** Marina Letica-Crepulja, Aleksandra Stevanović, Diana Palaić, Iva Vidović, Tanja Frančišković

**Affiliations:** 1Department of Psychiatry and Psychological Medicine, Faculty of Medicine, University of Rijeka, 51000 Rijeka, Croatia; aleksandras@medri.uniri.hr (A.S.); vidoviiva@gmail.com (I.V.); tanja.franciskovic@medri.uniri.hr (T.F.); 2Department of Psychiatry, Clinical Hospital Center Rijeka, Referral Center of the Ministry of Health of the Republic of Croatia, 51000 Rijeka, Croatia; diana.palaic@gmail.com; 3Department of Basic Medical Sciences, Faculty of Health Studies, University of Rijeka, 51000 Rijeka, Croatia

**Keywords:** COVID-19, PTSD, war-related stress, pandemic-related stress, treatment-seeking

## Abstract

Background: The aim of this study was to examine post-traumatic stress disorder (PTSD) symptom levels and coping strategies during the COVID-19 pandemic among treatment-seeking veterans with pre-existing PTSD. Method: A cohort of 176 male treatment-seeking veterans with pre-existing PTSD during the first COVID-19 pandemic lockdown (T1) and 132 participants from the same cohort one year after the onset of the pandemic (T2) participated in a longitudinal study. All participants responded to a COVID-19-related questionnaire and the following measures: the Life Events Checklist for DSM-5 (LEC-5), PTSD Checklist for DSM-5 (PCL-5) and the Brief COPE. Results: The intensity of overall PTSD symptoms, avoidance symptoms and negative alterations in cognitions and mood was lower at T2. PTSD symptoms were not significantly correlated with SARS-CoV-2 potentially traumatic events (PTE) at T2. Veterans scored higher on emotion-focused and problem-focused coping than on dysfunctional coping. Conclusions: Veterans with pre-existing PTSD who were receiving long-term treatment coped with COVID-19 stressors without the effects of retraumatization and a consequent worsening of PTSD symptoms.

## 1. Introduction

The coronavirus disease-19 (COVID-19) pandemic has caused multiple health, social, and economic stressors and presents a progressively emerging and potentially long-lasting life threat [1]. Pandemic stressors have a broad spectrum of impacts including mental health risk for individuals around the world. Extensive research has documented the negative psychiatric consequences of the COVID-19 pandemic [2,3,4]. A study that examined the global impacts of the pandemic on major depressive and anxiety disorders showed that prevalence of major depressive disorder and anxiety disorders had increased by more than 25 percent worldwide [3].

Patients with pre-existing mental disorders appeared to be at higher risk for wide-ranging mental health effects [5,6,7]. Regarding the veteran population, the results from recent studies revealed resilience to mental health problems and lower rates of suicidal ideation among US military veterans nearly 10 months into the pandemic [8,9]. However, these studies also showed that the prevalence of generalized anxiety disorder increased and the prevalence of major depressive disorder and post-traumatic stress disorder (PTSD) remained stable during the pandemic [10]. A recent longitudinal study, conducted during the period between two lockdowns among treatment-seeking veterans with pre-existing mental health difficulties in the UK, did not find any significant changes in symptoms of PTSD, common mental health disorders, anger, or alcohol use between the lockdowns [11]. A study that compared prepandemic and peripandemic levels of PTSD symptoms among Croatian treatment-seeking veterans with pre-existing PTSD demonstrated a reduction in PTSD symptom levels during the onset of the COVID-19 pandemic [12]. As the COVID-19 pandemic is often referred to as a “marathon, not a sprint,” and is spread around the world, there remains a need for further longitudinal studies to examine long-term changes in mental health of veterans with pre-existing difficulties in different settings.

Coping reflects a form of adaptation elicited by stressful circumstances. The previous research has repeatedly shown that combat veterans with PTSD use a non-adaptive and avoidance coping style [13,14,15]. The coping strategies veterans use when facing the pandemic stressors are important, because they can either alleviate or cause additional stress. There is a lack of systematic research on coping with pandemic stressors among veterans with pre-existing mental health difficulties including PTSD.

Almost thirty years after the Homeland War in Croatia (1991–1995), veterans still suffer from numerous health problems. In 2020, there were 33,089 treatment-seeking veterans with PTSD, which is 7.76% of the overall veteran population in Croatia [16]. A recent study revealed high rates of overall symptoms and greater severity of post-traumatic symptoms (i.e., complex PTSD) among treatment-seeking veterans years after the war ended [17]. The aim of this study was to follow the trajectories of PTSD symptom levels and examine coping strategies during the onset and one year after the onset of the COVID-19 pandemic among treatment-seeking veterans with pre-existing PTSD.

## 2. Materials and Methods

### 2.1. Participants and Procedure

All participants were male veterans from the Homeland War in Croatia (1991–1995) and had been in treatment for war-related PTSD for an average of 18.8 years (range 3 to 30 years at the second measurement) at the Referral Center of the Ministry of Health of the Republic of Croatia (RCPTSD) at the Clinical Hospital Center (CHC) Rijeka. Of 250 treatment-seeking veterans whom we approached, 176 (70.4%) participated at the first measurement (T1), which took place from April 15 to the end of May 2020 (i.e., during the onset of the COVID-19 pandemic). Sixty-four veterans could not be reached, five refused to participate, and five were excluded from further analysis due to incomplete data. They were recontacted during April and May 2021 (T2) when the third wave of the COVID-19 pandemic reached its peak. One hundred thirty-two (132) responded and consented to participate in the second measurement. Of the 44 who did not take part in the second measurement, five refused to participate, one had died, and the remaining 39 could not be reached even after three phone calls. Since their first referral, and before T1, participants had been involved in one or more outpatient treatment options: intensive PTSD program for day-care hospital (78.2%), long-term psychotherapy (26.3%), low-level treatment groups such as PTSD Club (11.3%), and regular outpatient psychiatric appointments (89.5%). Every other participant had received inpatient care (49.6%) at some point during their treatment. At T1 and T2, and in between, participants had been treated in regular outpatient psychiatric appointments with no changes in the administration of regular medication.

Due to the ongoing COVID-19 restrictions, the assessments were conducted by telephone or face-to-face with participants who attended the check-up. After the participants had been given detailed information about the study, all participants provided their written informed consent. In cases when the assessments were made by telephone, the written informed consent was provided at the next regular onsite appointment. The two parts of the evaluation consisted of a structured clinical interview and self-report questionnaires. Besides sociodemographic items, the structured interview at T2 included additional questions on difficulties with specific aspects of the COVID-19 pandemic. The study was approved by the Ethical Committee of CHC Rijeka, ethical approval code 003-05/20-1/85.

### 2.2. Measures

The measures used in the study were the Life Events Checklist for DSM-5 (LEC-5), PTSD Checklist for DSM-5 (PCL-5) and the Brief COPE. At the second assessment, the participants were asked whether they had experienced any traumatic life event in the past twelve months since the lifetime traumatization was assessed at T1.

The Life Events Checklist for DSM-5 (LEC-5) was used to assess possible traumatic events experienced by participants in the past twelve months [18]. The self-report measure lists 16 traumatic events and an additional item indicating any other stressful event. For the study, the total score of lifetime trauma, calculated as the sum of traumatic events, ranged from 0 to 17. The checklist was reported to have good psychometric properties [18,19].

The PTSD Checklist for DSM-5 (PCL-5) with Criterion A is a self-report measure revised to match the adapted DSM-5 criteria for PTSD [20]. A provisional PTSD diagnosis can be made considering items rated 2 = moderately or higher according to the DSM-5 diagnostic rule (at least one B, one C, two D, and two E symptoms present). Symptom severity was calculated as the sum of all items (0–80) or as the sum within a specific cluster of symptoms. Authors reported the score of 33 or above to be indicative of probable PTSD diagnosis and was thus used as the cut-off score in this study. Validation studies showed excellent psychometric properties for evaluating PTSD [20,21,22,23]. PCL-5 showed good internal consistency in our study, with Cronbach alphas ranging from 0.72 to 0.85 for clusters and 0.90 for total PCL-5.

The Brief COPE is a 28-item multidimensional measure of coping strategies used for regulating cognitions and behaviors in response to stressors [24]. Fourteen two-item scales are rated on the four-point rating scales (1 = I have not been doing this at all to 4 = I have been doing this a lot). The score of each scale is the sum of two items, with the possible range of 0–8. Each scale can be viewed independently or as part of emotion-focused, problem-focused, or dysfunctional coping strategies [24,25]. The Brief COPE is reported to have good psychometric properties [24]. Good psychometric properties were also reported for the Croatian version of Brief COPE in the study with a large adult online sample [26]. The Cronbach alpha in our study was 0.81.

The COVID-19-related questionnaire was created for specifically this study, and consisted of three parts. The first part was related to severe acute respiratory syndrome coronavirus 2 (SARS-CoV-2) infection of the participants and their close relatives, severity of the COVID-19 symptoms, the need for hospitalization, the level of recovery and death of a close family member. The second part was related to the level of compliance to the government COVID-19 measures. The third part was related to COVID-related stressors: pandemic duration, pandemic uncertainty, media coverage/exposure, restricted family gatherings, restricted access to medical services, fear of COVID-19, physical distancing, financial burden, self-isolation, restricted access to domestic supplies, and other.

### 2.3. Data Analysis

The existing dataset from T1 and the data collected at T2 were used for the analysis. Descriptive statistics were used to present frequencies/percentages or means and standard deviations for parametric measures. Pearson’s chi-square test was used to test for differences between groups on categorical variables. In cases where the cells’ frequency was below five, the Yates correction was applied. Group differences on continuous variables were tested with *t*-tests or Mann–Whitney U tests when number of cases was <20. Differences between measurements on continuous variables were tested with t-tests for repeated measurement. Spearman’s correlation coefficient was applied to test for significant correlations. Statistical significance was set as *p* < 0.05. Statistical analysis was performed with Statistica software, version 12 (Dell Inc., Tulsa, OK, USA).

## 3. Results

### 3.1. Demographics and War-Related Characteristics

Of 176 male participants in the first measurement, 132 (75%) war veterans participated in the second measurement. The sociodemographic characteristics are presented in Table 1. The missing 44 participants and those who participated at T2 did not differ significantly on any of the sociodemographic characteristics or war-related measures. Therefore, the functional equivalence and representativeness of the sample allowed for further analysis.

Since the previous assessment, none of the participants changed their educational level, four (3%) relocated, and six (4.5%) had their marital status changed. In the last year, 16 (12.1%) participants had their employment status changed as 4 participants found employment, 1 lost his job, and 11 retired. Eighteen participants (13.6%) reported a change in their economic status with 16 (12.1%) reporting lower economic status and 2 (1.5%) better economic status. The majority of participants (*n* = 101, 76.5%) did not experience a potentially traumatic event (PTE) over the past twelve months, 28 (21.2%) reported one PTE, and 3 (2.3%) participants reported from two to six PTE in the last year (assessed by LEC).

### 3.2. COVID-19 Related Characteristics

Since the first assessment when only one participant had been infected, 14 (10.6%) participants were infected with SARS-CoV-2 between T1 and T2. Five participants had mild, four had moderate and four had severe COVID-19 symptoms. One participant with severe COVID-19 symptoms had been treated in the intensive care unit. Eight had wholly recovered, and six reported they had partially recovered from COVID-19. Compared to the previous year when only 2 participants had a close family member who tested positive, at the second measurement, 46 (34.8%) reported a family member or a close person with COVID-19, and 5 participants had lost a close relative due to the illness.

The majority of the participants said that they had entirely followed (*n* = 71, 53.8%) or mostly followed (*n* = 41, 31.1%) the precautionary measures introduced by the government during the past year.

The participants were asked to assess if they were affected by COVID-19-related non-PTE stressors. The most challenging COVID-19-related non-PTE stressful aspect was pandemic duration, since 99 (75%) of participants reported to be affected by it. Other stressors were pandemic uncertainty (69.7%), media coverage/exposure to pandemic (67.4%) and restriction of family gatherings (65.9%). The least frequent COVID-19-related non-PTE stressor was self-isolation (35.6%) (Figure 1). The increase in number of experienced COVID-19-related non-PTE stressors was not significantly related to the overall PTSD symptom intensity at T2 (r = 0.15, *p* = 0.08).

### 3.3. PTSD Symptom Severity

Most of the participants scored 33 or higher on total symptom severity, indicative of probable PTSD diagnosis (T1 83.8%, T2 80.6%). The intensity of the overall PTSD symptoms, avoidance symptoms and negative alterations in cognitions and mood was significantly lower at T2. The change in intrusion symptoms and alterations in arousal and reactivity was not statistically significant (Table 2).

The PTSD symptoms were neither significantly correlated with SARS-CoV-2 (personally or a close one’s) infection nor the intensity of the COVID-19 symptoms (personally or of a close one) at T2. Of 37 participants with at least one PTE in the last year assessed by LEC, 9 experienced COVID-19-related PTE. A comparison between participants with COVID-19-related PTE (severe personal COVID-19 illness and death of a close one due to COVID-19) and participants with at least one other PTE in the last 12 months was made. The participants with COVID-19-related PTE (*n* = 9) did not differ in the intensity of overall PTSD symptoms at T2 compared to participants with other PTE (*n* = 28) in the last 12 months (U = 101.5, *p* = 0.39). In the overall sample, the number of experienced PTEs in the last year was not significantly related to overall PTSD symptom intensity (r = 0.12, *p* = 0.19).

The overall PTSD symptoms and cluster D symptoms were significantly correlated with perceived difficulties in dealing with media coverage (overall symptoms: r = 0.19, *p* < 0.05; cluster D symptoms: r = 0.23, *p* < 0.05) and with restricted access to domestic supplies (overall symptoms: r = 0.22, *p* < 0.05; cluster D symptoms: r = 0.24, *p* < 0.05). The increase in number of experienced COVID-19-related non-PTE was not significantly related to the overall PTSD symptom intensity at T2 (r = 0.15, *p* = 0.08).

### 3.4. Coping with COVID-19-Related Issues

In general, the average scores of individual coping strategies had diminished over a year. The average means for individual coping strategies in two measurements are presented in Figure 2. Acceptance and self-distraction continued to be the most frequently used coping strategies one year into the pandemic. The participants reported a significant decrease in self-distraction (t = 6.402, *p* < 0.001), active coping (t = 3.788, *p* < 0.001), denial (t = 2.825, *p* = 0.005), emotional support (t = 5.368, *p* < 0.001), venting (t = 3.489, *p* = 0.001), positive reframing (t = 3.821, *p* < 0.001), planning (t = 2.815, *p* = 0.006), and religion (t = 2.55, *p* = 0.012).

In an alternative way of grouping coping strategies, in the second measurement, veterans scored higher on emotion-focused coping (Mean = 4.07, SD = 1.01) and problem-focused coping (Mean = 3.71, SD = 1.57) than on dysfunctional coping (Mean = 3.13, SD = 0.96). Compared to the first measurement, there was a significant decrease in all three types of coping (emotion-focused coping: t = 4.72, *p* < 0.001; problem-focused t = 3.589, *p* < 0.001; dysfunctional coping: t = 3.45, *p* < 0.001) (Table 3).

## 4. Discussion

The aim of this study was to compare PTSD symptom levels and examine coping strategies at the onset and one year after the onset of the COVID-19 pandemic among treatment-seeking male veterans with pre-existing PTSD. The main finding was that the intensity of the overall PTSD symptoms (T1= 47.28, SD = 12.81; T2= 43.82, SD = 14.46; *p* = 0.016), avoidance symptoms and negative alterations in cognitions and mood decreased despite the greater exposure to COVID-19-related PTEs and the twelve-month longer exposure to COVID-19-related non-PTE stressors. The findings suggested that veterans with PTSD receiving a long-term treatment experienced COVID-19 stressors without the effects of retraumatization and a consequent worsening of PTSD symptoms. Stressful experiences to which the participants were exposed during the lockdown differ from the initial traumatic war-related stressors. Therefore, they had a lower potential for retraumatization (i.e., PTSD exacerbation after experiencing a new traumatic event). War-related stressors are central in the personal life experience of the participants and are the most substantial factors associated with PTSD [27,28,29]. The stressful experiences related to the COVID-19 pandemic appeared to be far from the central negative event to the participants’ life and identity. The findings of this study are consistent with emerging results regarding the mental health of veteran populations during the pandemic. In a nationally representative sample of US military veterans, the prevalence of major depressive disorder and PTSD positive screens remained stable while the prevalence of generalized anxiety disorder positive screens increased [10] and the rate of suicidal ideation decreased nearly 10 months into the pandemic [9]. Regarding the mental health among treatment-seeking veterans with pre-existing mental health difficulties, the UK study revealed no significant changes in symptoms of PTSD during the pandemic [11] and a study conducted in Croatia revealed a decrease in PTSD symptoms during the onset of the pandemic as compared to the measurement a year before [12]. PTSD symptoms and particularly avoidance symptoms may have reduced because of the restrictive measures that prevented greater extent of exposure to reminders of trauma in everyday life. Adherence to treatment, regularity of appointments, and stability of treatment as all participants continued to be treated as usual (TAU) may also be viewed as a functional coping strategy that prevented considerable risk of worsening PTSD symptoms [12]. The results emphasize the importance of treatment engagement of veterans who meet criteria for mental disorder, since approximately 60% of them do not seek help because of concerns about stigma, with many expecting to face prejudice and discrimination [30]. Appropriate treatment options during the COVID-19 pandemic to prevent, treat, and mitigate the effects of COVID-19 are likely to promote mental health and prevent its deterioration [5].

PTSD symptoms were neither significantly correlated with exposure to the COVID-19-related PTEs nor were they significantly correlated with the number of experienced PTEs between the two pandemic timepoints. Regarding COVID-19-related non-PTEs in general, the higher number of experienced stressors was not significantly related to the overall PTSD symptom intensity at the second measurement (r = 0.15, *p* = 0.08). Recent research on the correlation between pandemic-related stressors and mental health problems in the general population revealed higher levels of symptoms of adjustment disorder in those with a variety of COVID-19-related stressors [31]. Among veterans with an increased level of mental health suffering, different COVID-related stressors appeared to be the strongest risk factors for increased suicidality [9], increased distress [10], common mental disorders, and hazardous drinking [32]. In treatment-seeking veterans, those who reported more COVID-related stressors and lower levels of social support may have been particularly vulnerable to an increasing severity of a range of mental health difficulties [11,33]. The results of this study indicating no correlation of PTSD symptom level with COVID-related PTE and non-PTE can be explained by a lower potential of COVID-related stressful experiences worsening PTSD symptoms because they differ from the central negative traumatic events (war-related criterion A) [12,29]. As the veteran cohort in this study was aged 54 (SD = 5.97), the results could be explained in light of the recent findings indicating that older adults may be more resilient and less affected by the mental health effects of the pandemic than younger age groups [3].

The veterans used coping strategies to a lesser extent one year after than during the onset of the pandemic, which points to a certain level of adjustment to “the new normal” circumstances. Moreover, they used emotion-focused and problem-focused strategies, which prevented worsening of their PTSD symptoms and enabled better adjustment to the pandemic circumstances. Current literature suggests that certain coping strategies, such as avoidance coping, are associated with a greater PTSD severity [34,35] and others, such as problem- or emotion-focused coping strategies without the aid of social support, are protective and associated with lower levels of PTSD [36]. These findings are in line with recent reports from Canada, where veterans reported predominantly adaptive coping strategies in dealing with COVID-related stressors and dysfunctional strategies such as increased alcohol intake the least [37]. US veterans reported greater appreciation of life, closer interpersonal relationships, and an increased sense of personal strength due to coping with COVID-related stressors [8]. Most of the participants (*n* = 111, 84.8%) followed the protective measures introduced by the government during the pandemic. Compliance with the measures may be viewed as an adaptive coping strategy of acceptance. A possible explanation for compliance with the measures may be returning to “combat mode” and military subordination [12].

Despite the decreased intensity between the two measurements, the PTSD symptoms and symptoms of negative alterations in cognitions and mood were significantly correlated with perceived difficulties in dealing with media coverage (overall symptoms: r = 0.19, *p* < 0.05; negative alterations in cognitions and mood: r = 0.19, *p* < 0.05) and with restricted access to domestic supplies (overall symptoms: r = 0.22, *p* < 0.05; negative alterations in cognitions and mood r = 0.22, *p* < 0.05). It has been repeatedly emphasized after the exploration of the psychological, social, and neuroscientific effects of COVID-19 that one of the immediate mental health priorities is to establish longer-term strategies for mental health since intensive media consumption may amplify distress and anxiety, and optimal patterns of consumption may enhance wellbeing [38,39]. In research carried out on the general population, both participants who had been directly exposed to COVID-19 and participants who had been indirectly exposed to COVID-19 (e.g., via media) experienced PTSD-like symptoms [40], which points to a recommendation of minimizing COVID-related media consumption.

The current findings should be considered within the context of several limitations. The sample size was relatively small, which increases the risk of type II errors. The present findings may not be generalized to the wider population of treatment-seeking veterans with PTSD as the context of traumatization and the context of the pandemic were related to Croatia. Therefore, further research is needed to evaluate the generalizability of the current findings to female, younger or more diverse veteran samples.

## Figures and Tables

**Figure 1 jcm-11-02715-f001:**
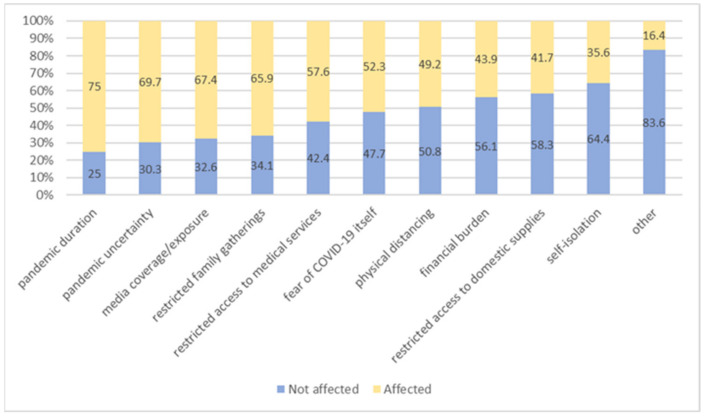
The proportion of the participants affected by COVID-19-related non-PTE stressors.

**Figure 2 jcm-11-02715-f002:**
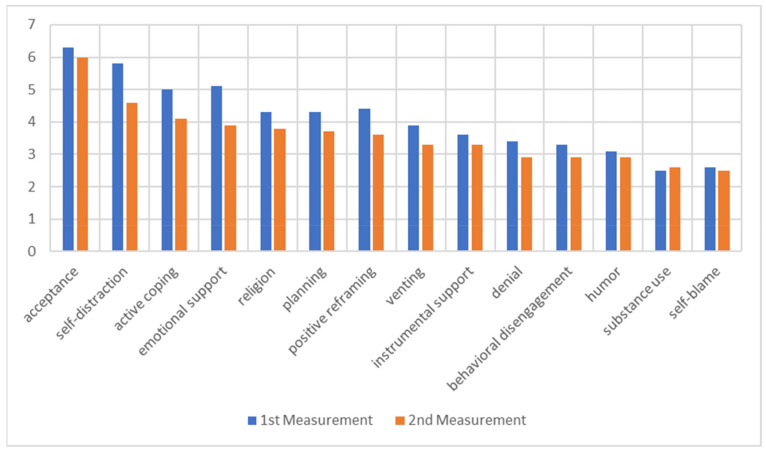
The average scores on individual coping strategies (the Brief COPE) in overall sample (range of 0–8).

**Table 1 jcm-11-02715-t001:** Sociodemographic characteristics of study participants at the first assessment (T1). Data are presented as count (percentage) unless otherwise indicated.

	The 1st Measurement Participants *n* = 176		Statistics	
	Participants Assessed at T1 and T2 *n* = 132	Participants Missing at T2 *n* = 44		
	X (SD)	X (SD)		
Age	53.38 (6.39)	54.16 (5.97)	t = −0.201	*p* = 0.113
	*n* (%)	*n* (%)		
Educational level				
Elementary school	11 (8.3)	4 (9.1)	χ^2^ = 0.042	*p* = 0.979
High school	108 (81.8)	36 (81.8)		
Higher education	13 (9.9)	4 (9.1)		
Work status				
Employed	34 (25.8)	6 (13.6)	χ^2^ = 4.658	*p* = 0.097
Unemployed	16 (12.1)	10 (22.7)		
Retired	82 (62.1)	28 (63.6)		
Marital status				
Married/cohabitating	93 (70.5)	36 (81.8)	χ^2^ = 2.767	*p* = 0.429
Single	20 (15.2)	5 (11.4)		
Divorced	16 (12.1)	2 (4.5)		
Other	3 (2.3)	1 (2.3)		
Economic status (self-reported)				
High	2 (1.5)	0 (0)	χ^2^ = 0.762	*p* = 0.683
Medium	83 (62.9)	27 (61.4)		
Low	47 (35.6)	17 (38.6)		
	X(SD)	X(SD)		
Treatment duration (in years)	17.8 (8.61)	15.71 (8.55)	t = 1.370	*p* = 0.173
Deployment duration (in months)	31.5 (19.43)	30.86 (22.73)	t = 0.164	*p* = 0.870
Life events (LEC-5)	10.04 (4.6)	9.44 (4.34)	t = 0.748	*p* = 0.455

**Table 2 jcm-11-02715-t002:** Overall PTSD symptom severity and PTSD symptom clusters severity on two measurements during the COVID-19 pandemic *.

	T1		T2			
	Range	Mean (SD)	Range	Mean (SD)	t	*p*
Cluster B symptoms (intrusion symptoms)	2–20	13.31 (4.05)	0–20	12.83 (4.60)	1.081	0.282
Cluster C symptoms (avoidance symptoms)	2–8	5.84 (1.78)	0–8	5.29 (2.24)	2.380	0.019
Cluster D symptoms (negative alterations in cognitions and mood)	1–29	16.83 (6.49)	0–32	15.45 (6.3)	1.998	0.048
Cluster E symptoms (alterations in arousal and reactivity)	2–24	13.18 (4.84)	2–24	12.41 (5.02)	1.440	0.152
PCL-5 total score	16–75	47.24 (12.87)	7–76	44.1 (14.09)	2.234	0.027

* Abbreviations: PTSD—post-traumatic stress disorder; SD—standard deviation; PCL-5—PTSD Checklist for DSM-5 (PCL-5) with Criterion A.

**Table 3 jcm-11-02715-t003:** Average scores for alternative grouping of the coping strategies on two measurements during the COVID-19 pandemic.

	T1	T2		
	Mean (SD)	Mean (SD)	t	*p*
Emotion focused	4.64 (1.06)	4.07 (1.01)	4.718	<0.001
Problem focused	4.31 (1.40)	3.71 (1.57)	3.589	<0.001
Dysfunctional	3.51 (0.87)	3.13 (0.96)	3.447	<0.001

## Data Availability

The data that support the findings of this study are available on doi: 10.5281/zenodo.5801404.

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
