# Peer review of "PTSD Symptoms and Coping with COVID-19 Pandemic among Treatment-Seeking Veterans: Prospective Cohort Study"

_jcm, 2022, doi:10.3390/jcm11102715_

Round 1

Reviewer 1 Report

Thanks for the opportunity to review this paper. I suggest a few ways to strengthen the article below:

  1. Introduction

This was reasonably well organised, though I thought the paragraph starting on line 47 could be more succinct. It appears that a few of the studies cited are longitudinal, though most seem to be cross sectional? This section could be strengthened by better outlining the need for longitudinal research specifically, and what has already been done in this line of work.

I suggest adding a section justifying why coping, as a concept, is important to examine, as well as any existing research in this area.

Later in the draft, it is clear that the research was conducted in Croatia. It would be helpful to have a section in the introduction why this is an important topic to examine in Croatia. Why is this geographic justified? Please add some detail about the Homeland war in Croatia which is referenced in the next section.

If the authors are struggling with word count limitation, I suggest making the paragraph starting on line 47 more succinct, and consider removing some of the text in the first paragraph unless it is crucial to the rest of the paper.

  1. Material and Methods

On line 78, I suggest specifying that they were ‘recontacted,’ as they were already contacted in 2020.

I suggest adding much more information on how participants were recruited. Phone calls were made, but how did researchers access their phone numbers in the first place? How was informed consent taken? Out of the number of veterans approached to take part, how many chose not to participate in the study?

Under ‘Measures,’ please add whether the checklists have been validated for Croatia. Also, please explain how the COVID 19 questionnaire was developed. Was it developed specifically by the authors for this study? Or was it developed elsewhere?

  1. Results

There seems to be a formatting issue on line 140 – two subheadings are side by side.

Under demographics, I expected to see a gender breakdown. Were all of the participants men? If so, clarify this in the text and add into the abstract as well.

There are some language/grammatical issues throughout the text. For instance, lines 186-7 make reference to ‘(personally or a close one).’ This is unclear. By close one, do the authors mean COVID 19 infection in a close family member?

The y axis is not labelled in Figure 2 (please check the other figures), and I could not tell from the text what the numbers referred to.

  1. Discussion

Please see lines 221-224. I suspect this needs to be removed.

The main finding is reasonably well contextualised within existing research. However, I think the authors could have offered up an explanation as to why their sample’s PTSD symptoms decreased during the pandemic.

What do the authors make of the fact that all types of coping (except substance use) were less frequent at T2 than at T1? On lines 274-276, it is stated that this may be due to adjusting to the new normal, but this explanation needs to be fleshed out a bit more.

  1. Conclusions

I believe the text in lines 328-9 should be removed.

I suggest summarising the main findings and strengths of the article.

Author Response

Dear Reviewer,

Thank you very much for your valuable comments and suggestions. The authors of the manuscript “PTSD symptoms and coping with COVID -19 pandemic among treatment-seeking veterans: prospective cohort study” would like to thank you for the effort and valuable input. We appreciate the opportunity to resubmit our article. We described how we have responded, point by point, to your comments. We provide the article with track changes and another copy with accepted changes. If you think there is anything else to be improved or we haven’t responded adequately to your comments, we stand available to do it.  

Best regards,

The authors of the manuscript

Point by Point Reply to Review Report (Reviewer 1)

Introduction

  1. This was reasonably well organised, though I thought the paragraph starting on line 47 could be more succinct. It appears that a few of the studies cited are longitudinal, though most seem to be cross sectional? This section could be strengthened by better outlining the need for longitudinal research specifically, and what has already been done in this line of work.

Ad 1. The suggestion was accepted, and the paragraph was improved in suggested manner (original manuscript with track changes, line 54-80).

  1. I suggest adding a section justifying why coping, as a concept, is important to examine, as well as any existing research in this area.

Ad 2. The paragraph related to coping was added (original manuscript with track changes, line 81-86).

  1. Later in the draft, it is clear that the research was conducted in Croatia. It would be helpful to have a section in the introduction why this is an important topic to examine in Croatia. Why is this geographic justified? Please add some detail about the Homeland war in Croatia which is referenced in the next section.

Ad 3. The clarification was provided  (original manuscript with track changes, line 89-93).

  1. If the authors are struggling with word count limitation, I suggest making the paragraph starting on line 47 more succinct and consider removing some of the text in the first paragraph unless it is crucial to the rest of the paper.

Ad 4.  The suggested removing of the text was conducted. (original manuscript with track changes, line 39-79).

Material and Methods

  1. On line 78, I suggest specifying that they were ‘recontacted,’ as they were already contacted in 2020.

Ad 5. The text was rephrased.  (original manuscript with track changes, line 116).

  1. I suggest adding much more information on how participants were recruited. Phone calls were made, but how did researchers access their phone numbers in the first place? How was informed consent taken? Out of the number of veterans approached to take part, how many chose not to participate in the study?

Ad 6.     The suggestion was accepted and explanation provided (original manuscript with track changes, line 107-127).

  1. Under ‘Measures,’ please add whether the checklists have been validated for Croatia. Also, please explain how the COVID 19 questionnaire was developed. Was it developed specifically by the authors for this study? Or was it developed elsewhere?

Ad 7.  The information is added (original manuscript with track changes  line 176  and 178-187)

Results

  1. There seems to be a formatting issue on line 140 – two subheadings are side by side.

Ad 8. The corrections were made.

  1. Under demographics, I expected to see a gender breakdown. Were all of the participants men? If so, clarify this in the text and add into the abstract as well.

Ad 9. The clarification were made (original manuscript with track changes, lines 19, 107, 210, 296)

  1. There are some language/grammatical issues throughout the text. For instance, lines make reference to ‘(personally or a close one).’ This is unclear. By close one, do the authors mean COVID 19 infection in a close family member?

Ad 10. The corrections were made (original manuscript with track changes, lines 180,182, 224, 225, 226, 253, 254, 257)

  1. The y axis is not labelled in Figure 2 (please check the other figures), and I could not tell from the text what the numbers referred to.

Ad 11.  Expelnation is provided in text  and Title of the Figure -range 0-8 (original manuscript with track changes, line 172 and 240)

Discussion

  1. Please see lines 221-224. I suspect this needs to be removed.

Ad 12. We apologize for the inconvenience.

  1. The main finding is reasonably well contextualised within existing research. However, I think the authors could have offered up an explanation as to why their sample’s PTSD symptoms decreased during the pandemic.

Ad 13. We agree with your observation. The aim was to follow the trajectories of PTSD symptom levels and examine coping strategies. There were expectations justified by the multiple expert opinions that COVID-19 pandemic and exposure to the related stressor will worsen the mental health among veterans particularly those with PTSD. That really concerned us and we tried not to have expectations of any kind and to tried preserve “naive eye” as much as possible during the research. With full acceptance of your comments mentioned under point 8. and 10. And the realistic insight in the results, particularly having clinical significance in our mind, we rephrased the Discussion section in the manner of no worsening during the pandemic rather than “improvement”.

  1. What do the authors make of the fact that all types of coping (except substance use) were less frequent at T2 than at T1? On lines 274-276, it is stated that this may be due to adjusting to the new normal, but this explanation needs to be fleshed out a bit more.

Ad 14.  We provided the explanation (original manuscript with track changes, line 365-368)

Conclusions

  1. I believe the text in lines 328-9 should be removed.

Ad 15. According to the recommendation of the Reviewer 3, we removed Conclusion section.

  1. I suggest summarising the main findings and strengths of the article.

Ad 16. According to the recommendation of the Reviewer 3, we removed Conclusion section.

Reviewer 2 Report

I appreciate the author's interest in this important topic.

A thorough copy edit for English language and grammar is required but of course this is not relevant to the scientific merit of the manuscript.

Abstract:

Needs an edit for language and clarity. The first sentence (Veterans have been repeatedly shown to be in increased risk of experiencing  social and psychological problems during emergencies)

is not correct and I suspect isn’t what the authors intend to say.

It sounds from the abstract as if the T1 and T2 samples are different individuals, though reading further it is clear that it they are the same.

Introduction:

There several misleading claims here. No serious scholars are advocating for the mere fact of the COVID pandemic to be considered a universal criterion A event, which is what the authors sound like they are saying. In addition, this claim:

“Veterans have been repeatedly shown to be in increased risk of experiencing social and psychological problems during emergencies as they often have high rates of pre-existing trauma and report high prevalence of psychiatric disorders”

Is not supported by the cited papers. The use of the term emergencies vague and problematic. If the authors are simply saying that previous trauma exposure increases risk for subsequent mental health symptoms, they should just say that.

The meaning of this sentence is not clear: “As longitudinal assessments of the same cohort over time are needed to understand whether mental health has worsened for individuals relative to their own mental health”

Participants:

It would be useful to understand a bit more about the treatment participants were receiving. Was this medication maintenance? Where symptoms in remittance? In other words, did they still meet diagnostic criteria for PTSD? 18 years seems like an extremely long time to be in treatment without symptom remittance.

 Results:

The PTSD numbers seem strange to me. It isn’t conceptually clear why we would expect any meaningful reduction of PTSD symptoms from T1 to T2 after an average of 18 years treatment prior to T1 with no change in treatment from T1-T2. The most plausible explanation for this seems to be regression to the mean. Furthermore, the reductions don’t seem to be clinically meaningful. A PCL-5 score of either 44 or 47 both meet the clinical cut point for a PTSD diagnosis, suggesting that participants are still highly symptomatic after long term treatment.

Discussion:

It seems like the first paragraph is the journal instructions for the discussion section? Also just a small issue of diction; aftermath and one year after sound like the same thing. I think the authors mean during onset and 1 year after?

Again, the main finding of decreased PTSD symptoms seems suspect to me. First, the reductions are extremely small, and while statistically significant do not seem to indicate any clinical significance. Second, why would such a change be an indicator of resilience when we have no evidence of previous change despite an average of 18 years of treatment? Are the authors saying that treatment doesn’t work but the pandemic did work to reduce chronic PTSD symptoms? In addition, the average PCL-5 scores at T1 and T2 were both indicative of a probable PTSD.

Author Response

Dear Reviewer,

Thank you very much for your valuable comments and suggestions. The authors of the manuscript “PTSD symptoms and coping with COVID -19 pandemic among treatment-seeking veterans: prospective cohort study” would like to thank you for the effort and valuable input. We appreciate the opportunity to resubmit our article. We described how we have responded, point by point, to your comments. We provide the article with track changes and another copy with accepted changes. If you think there is anything else to be improved or we haven’t responded adequately to your comments, we stand available to do it.  

Best regards,

The authors of the manuscript

Point by Point Reply to Review Report (Reviewer 2)

  1. A thorough copy edit for English language and grammar is required but of course this is not relevant to the scientific merit of the manuscript.

Ad 1. The paper has been proofread by an English language lector.

Abstract:

  1. Needs an edit for language and clarity. The first sentence (Veterans have been repeatedly shown to be in increased risk of experiencing social and psychological problems during emergencies) is not correct and I suspect isn’t what the authors intend to say.

Ad 2. The sentence was deleted and the paragraph was rephrased.

  1. It sounds from the abstract as if the T1 and T2 samples are different individuals, though reading further it is clear that it they are the same.

Ad 3. To emphasize the sameness of the participants we put that 176 treatment-seeking veterans with pre-existing PTSD during the first COVID-19 pandemic lockdown (T1) and 132 participants from the same cohort one year after the onset of the pandemic (T2) participated in a longitudinal study. (original manuscript with track changes, line 19-21)

Introduction:

  1. There several misleading claims here. No serious scholars are advocating for the mere fact of the COVID pandemic to be considered a universal criterion A event, which is what the authors sound like they are saying.

Ad 4.  We accept the observation. Consequently, we deleted the related content. The content possibly had resulted from our own concern due to tendency of trivialization of the concept of psychotrauma and overexploitation of the term in media.  

Introduction:

  1. In addition, this claim: “Veterans have been repeatedly shown to be in increased risk of experiencing social and psychological problems during emergencies as they often have high rates of pre-existing trauma and report high prevalence of psychiatric disorders”. Is not supported by the cited papers. The use of the term emergencies vague and problematic. If the authors are simply saying that previous trauma exposure increases risk for subsequent mental health symptoms, they should just say that.

Ad 5.  The sentence was deleted, and the subsequent part of the paragraph was rephrased. (original manuscript with track changes line 37-53)

  1. The meaning of this sentence is not clear: “As longitudinal assessments of the same cohort over time are needed to understand whether mental health has worsened for individuals relative to their own mental health”

Ad 6. The above-mentioned sentence was deleted and the sentence: „ However, there remains a need for further longitu-dinal studies to examine changes in mental health of veterans with pre-existing difficulties during the COVID-19 pandemic. “was added. (original manuscript with track changes, line 77-80) 

Participants:

  1. It would be useful to understand a bit more about the treatment participants were receiving. Was this medication maintenance? Where symptoms in remittance? In other words, did they still meet diagnostic criteria for PTSD? 18 years seems like an extremely long time to be in treatment without symptom remittance.

Ad 7. The treatment at T1 and T2 is explained (lines cca 121-128). War stressors are in the group of traumatic stressors with the highest risk for long-term consequences for mental health, sometimes even life-long suffering, and yes, treatment usually include medication maintenance and regular controls. The percentage of these patients in the overall veteran population is provided now in the Introduction. (original manuscript with track changes, line 89-97 ). The levels of the symptoms are provided in the Result section (original manuscript with track changes 242-243)

 Results:

  1. The PTSD numbers seem strange to me. It isn’t conceptually clear why we would expect any meaningful reduction of PTSD symptoms from T1 to T2 after an average of 18 years treatment prior to T1 with no change in treatment from T1-T2. The most plausible explanation for this seems to be regression to the mean. Furthermore, the reductions don’t seem to be clinically meaningful. A PCL-5 score of either 44 or 47 both meet the clinical cut point for a PTSD diagnosis, suggesting that participants are still highly symptomatic after long term treatment.

Ad 8. We agree with your observation. The aim was to follow the trajectories of PTSD symptom levels and examine coping strategies. There were expectations justified by the multiple expert opinions that COVID-19 pandemic and exposure to the related stressor will worsen the mental health among veterans particularly those with PTSD. That really concerned us and we tried not to have expectations of any kind and to tried preserve “naive eye” as much as possible during the research. With full acceptance of your comments mentioned under point 8. and 10. And the realistic insight in the results, particularly having clinical significance in our mind, we rephrased the Discussion section in the manner of no worsening during the pandemic rather than “improvement”.

Discussion:

  1. It seems like the first paragraph is the journal instructions for the discussion section?

Ad 9. We apologize for the inconvenience.

  1. Also just a small issue of diction; aftermath and one year after sound like the same thing. I think the authors mean during onset and 1 year after?

Ad 9. Corrections were made throughout the text.

  1. Again, the main finding of decreased PTSD symptoms seems suspect to me. First, the reductions are extremely small, and while statistically significant do not seem to indicate any clinical significance. Second, why would such a change be an indicator of resilience when we have no evidence of previous change despite an average of 18 years of treatment? Are the authors saying that treatment doesn’t work but the pandemic did work to reduce chronic PTSD symptoms? In addition, the average PCL-5 scores at T1 and T2 were both indicative of a probable PTSD.

Ad 10. Refer to point Ad 8.

Reviewer 3 Report

This is strong paper and I only have some minor comments

Results

1. Very minor point but helpful to name the PCL-5 (PTSD_) symptom clusters rather than just say cluster C - e.g. Avoidance, Re-experiencing etc as this improves the clarity of the paper (also to do this in the discussion)

Discussion

1. 1st Paragraph looks like a note to the authors and should be removed.  "Authors should discuss the results and how they can be interpreted from the perspective of previous studies and of the working hypotheses. The findings and their implications should be discussed in the broadest context possible. Future research directions may also be highlighted"

2. It may be helpful to tone down the language as it could be 'considerable resilience' or it could be other factors such as natural improvements in PTSD symptoms, or veterans being in lockdown meant they didn't have to leave their homes so were not being exposed to potential triggers?

3. Am not sure what this sentence means 'A lack of longitudinal research worldwide has been subsequently overcome by longitudinal assessments of the same veteran cohort'

4. I would remove this paragraph: 'Regarding the correlation with restricted access to domestic supplies with intensity of the PTSD symptoms, it can be explained by the massive disproportionate response to the actual threat manifested for example, in panic buying of essential consumer items that has let to artificially produced global shortages [43,47]. '

5. This seems to be an a note for the authors - please remove 'Conclusions This section is not mandatory but can be added to the manuscript if the discussion is unusually long or complex.'

Author Response

Dear Reviewer,

Thank you very much for your valuable comments and suggestions. The authors of the manuscript “PTSD symptoms and coping with COVID -19 pandemic among treatment-seeking veterans: prospective cohort study” would like to thank you for the effort and valuable input. We appreciate the opportunity to resubmit our article. We described how we have responded, point by point, to your comments. We provide the article with track changes and another copy with accepted changes. If you think there is anything else to be improved or we haven’t responded adequately to your comments, we stand available to do it.  

Best regards,

The authors of the manuscript

Point by Point Reply to Review Report (Reviewer 3)

Results

  1. Very minor point but helpful to name the PCL-5 (PTSD_) symptom clusters rather than just say cluster C - e.g. Avoidance, Re-experiencing etc as this improves the clarity of the paper (also to do this in the discussion)

Ad 1. The clarification was made (original manuscript with track changes,lines 243-245, 298)

Discussion

  1. 1st Paragraph looks like a note to the authors and should be removed. "Authors should discuss the results and how they can be interpreted from the perspective of previous studies and of the working hypotheses. The findings and their implications should be discussed in the broadest context possible. Future research directions may also be highlighted"

Ad 2. We apologize for the inconvenience.

  1. It may be helpful to tone down the language as it could be 'considerable resilience' or it could be other factors such as natural improvements in PTSD symptoms, or veterans being in lockdown meant they didn't have to leave their homes so were not being exposed to potential triggers?

Ad 3. We agree with your observation. The aim was to follow the trajectories of PTSD symptom levels and examine coping strategies. There were expectations justified by the multiple expert opinions that COVID-19 pandemic and exposure to the related stressor will worsen the mental health among veterans particularly those with PTSD. That really concerned us and we tried not to have expectations of any kind and to tried preserve “naive eye” as much as possible during the research. With full acceptance of your comments mentioned under point 8. and 10. And the realistic insight in the results, particularly having clinical significance in our mind, we rephrased the Discussion section in the manner of no worsening during the pandemic rather than “improvement”.

  1. Am not sure what this sentence means 'A lack of longitudinal research worldwide has been subsequently overcome by longitudinal assessments of the same veteran cohort'

Ad 4. The sentence was removed.  

  1. I would remove this paragraph: 'Regarding the correlation with restricted access to domestic supplies with intensity of the PTSD symptoms, it can be explained by the massive disproportionate response to the actual threat manifested for example, in panic buying of essential consumer items that has let to artificially produced global shortages [43,47]. '

Ad 5. The paragraph was removed.

  1. This seems to be an a note for the authors - please remove 'Conclusions This section is not mandatory but can be added to the manuscript if the discussion is unusually long or complex.'

Ad 6. The Conclusions were removed.

Reviewer 4 Report

The current study was a long-term examination of veterans with PTSD and their response to the COVID pandemic lockdowns. Overall, I think that this study is an interesting replication of previous studies suggesting that PTSD symptomology decreased during the pandemic for veterans. I only have relatively minor points that I think should be addressed:

The fact that this uses an existing dataset should be acknowledged.

The authors should discuss the possibility that PTSD symptoms may have reduced (particularly avoidance symptoms) because PTSD patients would not have had the opportunity to be in the outside world and be confronted with reminders of trauma during normal everyday life.

The authors should also include some more relevant information on the correlation between clinical symptoms and COVID pandemic in healthy participants in the introduction/discussion to try to explain the findings of the current study.

General points:

First sentence introduction – should be ‘presents’

Line 54 – should be mental disorders

Line 77- 80 – why is there need for more research on the topic? Not clear based on the previous paragraphs

Line 219 – are these the percentages of participants who had COVID at the time of T2 assessment, or is this the percentage of participants who had COVID at any point in between T1 and T2?

Line 282 – use Mean = rather than X = please for clarity

Author Response

Dear Reviewer,

Thank you very much for your valuable comments and suggestions. The authors of the manuscript “PTSD symptoms and coping with COVID -19 pandemic among treatment-seeking veterans: prospective cohort study” would like to thank you for the effort and valuable input. We appreciate the opportunity to resubmit our article. We described how we have responded, point by point, to your comments. We provide the article with track changes. If you think there is anything else to be improved or we haven’t responded adequately to your comments, we stand available to do it.  

Best regards,

The authors of the manuscript

Point by Point Reply to Review Report 

  1. The fact that this uses an existing dataset should be acknowledged.

Ad 1. The statement regarding the existing dataset and the data collected at T2 is provided in Line 190. The research is conducted first hand to obtain data at T2. The data obtained at T2 were used in no previous research. The functional equivalence and representativeness of the sample at T2 allowed for further analysis (Explanation provided Line 201-206).

  1. The authors should discuss the possibility that PTSD symptoms may have reduced (particularly avoidance symptoms) because PTSD patients would not have had the opportunity to be in the outside world and be confronted with reminders of trauma during normal everyday life.

Ad 2. We accepted the reviewer's suggestion and added the sentence related to the possible impact of the restrictive measures on the level of the symptoms (Line 327-329)

  1. The authors should also include some more relevant information on the correlation between clinical symptoms and COVID pandemic in healthy participants in the introduction/discussion to try to explain the findings of the current study.

Ad 3. We appreciate the reviewer’s comment, and we will try to explain why we find it hardly possible to make such a comparison. In our study we assessed the level of the war-related PTSD symptoms among veterans involved in long-term treatment. So, during the assessment, participants assessed the level of the symptoms having in mind the worst traumatic event, which for PCL questionnaire means the event that bothered them the most in the past month. For all of them the worst event was one of the war-related traumas. It is not the case in general population. So, in our opinion it would be very difficult to explain the differences in mental health consequences in different populations as we assessed only war-related PTSD symptoms in veteran population. We hope that we explained our point of view but as we already noted, we stand on the disposal to perform further clarifications if needed.

Form Line 37 to 39 we reflected on the global impact of the COVID-19 pandemic on mental health in general population.

General points:

  1. First sentence introduction – should be ‘presents’

Ad 4. The correction was made.

  1. Line 54 – should be mental disorders

Ad 5. The correction was made.

  1. Line 77- 80 – why is there need for more research on the topic? Not clear based on the previous paragraphs

Ad 6. We rephrased the sentence emphasising the importance of the long-term research among veterans from different backgrounds (Line 78 - 79).

  1. Line 219 – are these the percentages of participants who had COVID at the time of T2 assessment, or is this the percentage of participants who had COVID at any point in between T1 and T2?

Ad 7. The clarification is provided (Line 221)

  1. Line 282 – use Mean = rather than X = please for clarity

Ad 8 The corrections were made.